# Association between Breast Cancer and Second Primary Lung Cancer among the Female Population in Taiwan: A Nationwide Population-Based Cohort Study

**DOI:** 10.3390/cancers14122977

**Published:** 2022-06-16

**Authors:** Fan-Wen Lin, Ming-Hsin Yeh, Cheng-Li Lin, James Cheng-Chung Wei

**Affiliations:** 1Department of Surgery, Chung Shan Medical University Hospital, Taichung 40201, Taiwan; ginobsesses@hotmail.com.tw; 2Division of Breast Surgery, Department of Surgery, Chung Shan Medical University Hospital, Taichung 40201, Taiwan; 3Management Office for Health Data, China Medical University Hospital, Taichung 40202, Taiwan; orangechengli@gmail.com; 4College of Medicine, China Medical University, Taichung 40402, Taiwan; 5Department of Allergy, Immunology & Rheumatology, Chung Shan Medical University Hospital, Taichung 40201, Taiwan; wei3228@gmail.com; 6Institute of Medicine, College of Medicine, Chung Shan Medical University, Taichung 40201, Taiwan; 7Graduate Institute of Integrated Medicine, China Medical University, Taichung 40201, Taiwan

**Keywords:** breast cancer, lung cancer, Asian women

## Abstract

**Simple Summary:**

There is an increasing number of patients with breast cancer and second primary lung cancer clinically. The aim of our population-based cohort study was to investigate this correlation in Taiwanese women using the National Health Insurance Research Database from Taiwan National Health Insurance. We confirmed that patients with breast cancer had a significantly higher risk of developing second primary lung cancer compared with patients without breast cancer, particularly in younger groups and in those without any comorbidities. The special association is meant to raise awareness and provoke interest in routine lung cancer screening for female patients who were diagnosed with breast cancer at a relatively young age.

**Abstract:**

Purpose: A special association between breast cancer and second primary lung cancer in Taiwanese women has been discovered not only in clinical practice, but also in a large population-based study. We hereby investigate the association between breast cancer and second primary lung cancer in Taiwanese women. Methods: This study was conducted from the National Health Insurance Research Database (NHIRD) from Taiwan National Health Insurance (NHI). Patients older than 18 years old and hospitalized with a first diagnosis of breast cancer during 2000 to 2012 were enrolled in the breast cancer group. Patients who were cancer free were frequency-matched with the breast cancer group by age (every five-year span) and index year. The ratio of breast cancer group to non-breast cancer group was 1:4. The event as the outcome in this study was lung cancer. The comorbidities viewed as important confounding factors included coronary artery disease, stroke, hypertension, diabetes, chronic obstructive pulmonary disease, hyperlipidemia, tuberculosis, chronic kidney disease, and chronic liver disease and cirrhosis. We estimated the hazard ratios (HRs), adjusted hazard ratios (aHRs), and 95% confidence intervals (CIs) for risk of lung cancer in the breast cancer group and non-breast cancer group using Cox proportional hazard models. Sensitivity analysis was also done using propensity score matching. All of the statistical analyses were performed using SAS statistical software, version 9.4 (SAS Institute Inc., Cary, NC). Results: There were 94,451 breast cancer patients in the breast cancer group and 377,804 patients in the non-breast cancer group in this study. After being stratified by age, urbanization level, and comorbidities, the patients with breast cancer had a significantly higher risk of lung cancer compared with the patients without breast cancer, particularly for those who aged between 20 and 49 years (aHR = 2.10, 95% CI = 1.71–2.58), 50 and 64 years (aHR = 1.35, 95% CI = 1.15–1.58), and those without any comorbidities (aHR = 1.92, 95% CI = 1.64–2.23). Conclusion: Patients with breast cancer had a significantly higher risk of developing second primary lung cancer compared with patients without breast cancer, particularly in younger groups and in those without any comorbidities. The special association may be attributed to some potential risk factors such as genetic susceptibility and long-term exposure to PM2.5, and is supposed to increase public awareness. Further studies are necessary given the fact that inherited genotypes, different subtypes of breast cancer and lung cancer, and other unrecognized etiologies may play vital roles in both cancers’ development.

## 1. Introduction

The incidences of breast cancer and lung cancer among the female population has been increasing for years thanks to well-established cancer screening systems and various high-quality handy diagnostic tools. In 2017, the incidence of female breast cancer and female lung cancer was 78.9 per 100,000 population and 31.6 per 100,000 population, as claimed by the data from the Health Promotion Administration, Ministry of Health and Welfare of Taiwan. It has become a public health burden, with increasing numbers of women suffering from both cancers.

According to our clinical experience, second primary lung cancer is frequently seen in women with breast cancer. Identifying the association can, in a way, offer an insight into the early prevention of a second primary malignancy. The issue has been brought up in some studies; however, none of which have targeted Asian population until 2018. Given the genetic variation for specific cancers and environmental differences between Asian and non-Asian ethnicities, Lin et al. conducted the first Asian population-based cohort study to report a special association between primary lung cancer and breast cancer in Taiwanese women by utilizing the data of the Taiwan Cancer Registry (TCR) and National Health Insurance (NHI). They reached the conclusion that lung cancer is associated with an increased risk of synchronous breast cancer in Taiwan, and vice versa [1]. However, the interference of some covariates as potential confounding factors was not discussed. For example, comorbidities and lifestyles were not involved in their study. Moreover, the association between lung cancer and breast cancer was not identified when two primary cancers were diagnosed more than six months apart because they initially defined synchronous malignancy as when two types of primary cancer were diagnosed within a six-month period. It must be stated that lung cancer is often presented as a second primary malignancy among breast cancer patients after years of follow-up clinically, which might not be compatible with the synchronicity emphasized by previous studies.

The aim of our study is to investigate the association between breast cancer and second primary lung cancer in Taiwanese women, as we hypothesized that patients with breast cancer have a higher risk of developing second primary lung cancer after a period of no less than six months since the breast cancer was first diagnosed.

## 2. Materials and Methods

### 2.1. Data Sources

This retrospective study was conducted from the National Health Insurance Research Database (NHIRD), which is the database of the comprehensive claim data from the Taiwan National Health Insurance program (Taiwan NHI). This government-based health insurance data cover over 99% of 23 million Taiwan citizens, and include the outpatients, hospitalization, medications records, and other medical services. We used all hospitalization files to conduct the analyses. All of the identification numbers were removed before the database was released. The history diagnoses were coded according to the International Classification of Disease, Ninth Revision, Clinical Modification (ICD-9-CM). The Research Ethics Committee of China Medical University and Hospital in Taiwan approved the study (CMUH-104-REC2-115-R6).

Taiwan launched their National Health Insurance (NHI) in 1995, operated by a single-buyer, the government. Medical reimbursement specialists and peer review scrutinize all of the insurance claims. The diagnoses of disease are based on ICD-9 codes, which are judged and determined by related specialists and physicians according to the standard imaging and clinical criteria. Therefore, the diagnoses and codes for diseases used in this cohort study should be correct and reliable.

Regarding disease definition and register, breast cancer and lung cancer are classified as a catastrophic illness in the NHI system in Taiwan. People who are diagnosed with breast cancer or lung cancer have the right to apply for a special “catastrophic illness card” and can benefit by receiving a medical discount against this disease. Consequently, there is a conscientious and careful process for distributing the “catastrophic illness card”.

Regarding the validity of cancer diagnoses in the NHI database, the positive predictive value (PPV) of the NHI database cancer diagnoses is 94% for all cancers. The PPV of lung cancer and female breast cancer is 95% and 92%, respectively [2].

### 2.2. Study Population

Patients older than 18 years old and hospitalized with the first diagnosis of breast cancer (ICD-9-CM 174) during 2000 to 2012 were enrolled in the breast cancer group. Patients who were cancer free were frequency matched with the breast cancer group by age (every five-year span) and index year. The ratio of breast cancer group to non-breast cancer group was 1:4 and the index date was set up as the first diagnosis date of breast cancer.

Both cohorts were followed up until a new diagnosis of lung cancer, based on the catastrophic illness card records. Individuals were censored at death, loss of follow-up, withdrawal from the insurance system, or the end of 2013, which ever came first.

### 2.3. Outcome and Co-Variates

The event as the outcome in this retrospective study was lung cancer (ICD-9-CM 162).

The exclusion criteria were patients younger than 18 years, male, with other cancer before index date, and being diagnosed with lung cancer within 6 months from the index date. Considering that comorbidities were important confounding factors, we defined the individuals with a history of comorbidities before the index date and who had at least once hospitalization record. The comorbidities included coronary artery disease (ICD-9-CM 410–414), stroke (ICD-9-CM 430–438), hypertension (ICD-9-CM 401-405), diabetes (ICD-9-CM 250), chronic obstructive pulmonary disease (ICD-9-CM 490, 492, 496), hyperlipidemia (ICD-9-CM 272), tuberculosis (ICD-9-CM 01), chronic kidney disease (ICD-9-CM 585), and chronic liver disease and cirrhosis (ICD-9-CM 571). Urbanization level instead of geographical area was also used to address the environmental issues, which represents the differences in the population density and socioeconomic status between different areas.

### 2.4. Statistical Analysis

In this retrospective study, we presented continuous variables by mean and standard deviation, and categorical variables by number and percentage. The differences between the breast cancer group and non-breast cancer group were estimated using the Chi-square test and t-test in the continuous and categorical variables, respectively. The incidence rate of lung cancer was calculated for both cohorts, and was estimated as the number of lung cancer occurrence divided by follow-up time (per 10,000 person-years). The Kaplan−Meier method was used to measure the cumulative incidence curves for each cohort and the log rank test was applied to assess the difference between two survival curves. We estimated crude hazard ratios (HRs), adjusted hazard ratios (aHRs), and 95% confidence intervals (CIs) for risk of lung cancer in the breast cancer group and non-breast cancer group using Cox proportional hazard models. Sensitivity analysis was also done using propensity score matching. Patterns of lung cancer incidence in breast cancer were compared with those of the general population using standardized incidence ratios (SIRs). SIR was calculated as the number of observed lung cancer cases among breast cancer divided by the excepted number of lung cancer cases. The number of lung cancer cases was obtained from the product of the national age-specific, gender-specific incidence rates obtained from the Registry of Catastrophic Illness Patient Database (RCIPD). All of the statistical analyses were performed using SAS statistical software, version 9.4 (SAS Institute Inc., Cary, NC, USA). The figure of the cumulative incidence curve was plotted using R software. The significant level set at less than 0.05 for two-side testing of the *p*-value.

## 3. Results

To clarify the association between breast cancer and lung cancer, Table 1 shows the number of patients for each variable in two cohorts. There were 94,451 breast cancer patients and 377,804 patients in the non-breast cancer group in this study, and the mean age in the breast cancer group and non-breast cancer group was 52.7 years old. Geographical distribution showed no remarkable difference between both cohorts. The breast cancer patients with stroke and COPD had a significantly lower percentage than the non-breast cancer group (*p* < 0.001 and *p* = 0.01, respectively); however, patients with breast cancer had a higher percentage of hyperlipidemia, hypertension, diabetes, and chronic liver disease and cirrhosis as comorbidities (all *p* < 0.001). The initially matched analysis was added in the right column as a supplement.

Figure 1 demonstrates that patients with breast cancer had significantly higher cumulative incidence of lung cancer than patients with non-breast cancer (*p* < 0.001).

Table 2 presents the incidence and risk factors of lung cancer. The incidence of lung cancer was 8.20 and 5.94 per 10,000 person-years in the breast cancer group and non-breast cancer group, respectively. After adjusting for age, urbanization level, and comorbidities, patients with breast cancer had a significantly higher risk of developing lung cancer (aHR = 1.34, 95% CI = 1.20–1.49) compared with patients without breast cancer. With increasing age, patients aged 50 to 64 years (aHR = 2.41, 95% CI = 2.13–2.71) and more than 65 years old (aHR = 4.17, 95% CI = 3.63–4.79) had a significantly higher risk of lung cancer compared with those aged 20 to 49 years old. Patients with COPD (aHR = 1.68, 95% CI = 1.37–2.08) and hypertension (aHR = 1.97, 95% CI = 1.76–2.22) had a significantly higher risk of developing lung cancer compared with patients with CAD (aHR = 0.68, 95% CI = 0.57–0.81) and chronic liver disease and cirrhosis (aHR = 0.63, 95% CI = 0.45–0.90) as comorbidities.

After being stratified by age and comorbidities (Table 3), patients between 20 and 49 years old (aHR = 2.11, 95% CI = 1.72–2.59) and 50 and 64 years old (aHR = 1.34, 95% CI = 1.14–1.58) and without any comorbidities (aHR = 1.94, 95% CI = 1.66–2.26), compared with non-breast cancer patients, patients with breast cancer had a significantly higher risk of lung cancer after being adjusted for age and comorbidities. Interestingly, the risk of developing second primary lung cancer among breast cancer patients did not appear to have a linear increase with a longer follow-up period.

The sensitivity analysis also showed that patients with breast cancer had a higher risk of developing lung cancer after being adjusted for age and multiple comorbidities (aHR = 1.21, 95% CI = 1.06–1.38) (Table 4).

Additional information with a flowchart of the study population inclusion is shown in Appendix A. The patterns of lung cancer incidence in breast cancer were compared with those of the general population using standardized incidence ratios (SIRs), as demonstrated in Appendix A.

## 4. Discussion

The nationwide population-based cohort study was conducted based on the National Health Insurance Research Database (NHIRD) from the Taiwan National Health Insurance (NHI). A total number of 472,255 patients, with 94,451 breast cancer patients in the breast cancer group and 377,804 non-breast cancer patients in the non-breast cancer group were enrolled in order to clarify the association between breast cancer and second primary lung malignancy.

After adjusting for age, urbanization levels, and comorbidities, patients with breast cancer had a significantly higher risk of developing lung cancer compared with patients without breast cancer, particularly in younger groups and in those without any comorbidities. The reason the correlation between breast cancer and second primary lung cancer was not discovered in elder or comorbid groups may be explained by the fact that aging and many underlying systemic diseases are risk factors of both cancers themselves. The correlation could be attenuated by these interference factors. Moreover, the result from our study also indicated a longer follow-up period might not be a major risk factor of developing second primary lung cancer among breast cancer patients.

A population-based cohort study conducted by Lin et al., as the first Taiwanese population-based cohort study to report a special association between primary lung cancer and breast cancer in women, highlighted the importance of synchronicity in double lung cancer/breast cancer, and suggested that radiation exposure is very unlikely to be a major risk factor for lung cancer in breast cancer survivors in Taiwanese women. In addition, the study found an interesting trend of an inverse correlation between the risk of second primary lung cancer and the age of breast cancer onset, although statistical significance was not reached. An increased risk for synchronous lung cancer was also observed in patients with HER2-positive breast cancer. They concluded that an inherited genetic background may play a vital role in disease phenotypes [1].

Almost the same results were noticed in our study, as we disclosed the association between breast cancer and second primary lung cancer, especially in younger groups and in those without comorbidities.

Nonetheless, there were three major differences between our study and Lin et al.’s one: First, we were unable to identify the importance of “synchronicity” in double lung cancer/breast cancer as a consequence of different settings in our study. To be better clarified, the previous study defined “synchronous malignancy” as when two types of primary cancer were diagnosed within a six-month period. They reached the conclusion that lung cancer is associated with an increased risk for synchronous breast cancer in Taiwanese women and vice versa. Instead, we were more interested in and focused on the association between the two cancers beyond six-month intervals; thus, we hypothesized that patients with breast cancer might have a higher risk of developing second primary lung cancer after a period of no less than six months since breast cancer was first diagnosed. Second, the breast cancer subtypes were not evaluated in our study because of the limited information about the diagnosis codes. A more practicable method was to categorize each subtype based on the treatment used and documented in NHI. For examples, Lin et al. defined breast cancer as human epidermal growth factor receptor 2 (HER2) positive if anti-HER2 targeted therapy was prescribed, and breast cancer was defined as hormone receptor positive if hormone therapy was recorded. However, this could be somehow inaccurate considering the guideline of NHI reimbursement was restrictive. The cost of targeted therapy was too expensive for many patients to afford at times, and self-paid drug prescriptions are not included in NHI. Third, multiple comorbidities, including CAD, stroke, hypertension, diabetes, COPD, hyperlipidemia, tuberculosis, chronic kidney disease, and chronic liver disease and cirrhosis, were deemed as important confounding factors in our study, none of which were considered in Lin et al.’s study.

The correlation between lung cancer and breast cancer has been elucidated by some established theories, as most of the literature, including that of Lin et al. and other ongoing in vitro studies, have attributed the association to inherited susceptibility [1,3,4,5,6,7,8,9]. After searching and reviewing some of the research literature, we supposed that air pollution, or more specifically, long-term exposure to fine particulate matter (PM2.5), may be another risk factor that leads to the carcinogenesis of the two cancers. PM2.5 is a well-known endocrine disruptor, which is composed of sulfate, nitrate, ammonium, elemental carbon, organic carbon, silicon, sodium ion, etc. The analyses regarding PM2.5 compositions from different regions generally emphasize the presence of phthalates in PM2.5. It must be pointed out that both PM2.5 and phthalates are believed to cause lung cancer and breast cancer, according to several studies [10,11,12,13].

Aside from genetics and PM2.5 exposure, we considered different lifestyles [14], viral infections [15,16,17,18,19,20,21], hormone [22], long-term night-shift work [23,24,25], and some undiscovered exposures may be potential risk factors promoting both lung cancer and breast cancer. Studies regarding these topics remain somehow limited and thus require further exploration.

There were some strengths in our study design. Firstly, this study was a population-based cohort study on a national scale conducted from the National Health Insurance Research Database (NHIRD), with arguably one of the largest case numbers of breast cancer in Asian women. The robust sample size was expected to scale down the sampling error. Secondly, we managed to minimize the confounders’ impact, thereby the main result was adjusted for several underlying comorbidities and age. The urbanization level was also utilized to address the environmental issues. Finally, the NHI has established strict guidelines for cancer diagnosis, which ensured the accuracy of our original data.

Several limitations must be recognized in this study. First, the influence of lifestyle, as a possible confounder, was not evaluated in a clear and precise way. While information of lifestyles was difficult to obtain, we assumed that it could be partially compensated by making the urbanization levels and comorbidities as a proxy. It should be acknowledged that we did not have information on smoking, which is a major risk factor of lung cancer. Instead, we used several comorbidities, including CAD, stroke, and COPD, for adjustment so as to minimize the influence of smoking. Secondly, the information regarding subtypes of both breast cancer and lung cancer were also lacking in our study. It was unclear whether a subtype of breast cancer was related to a certain histological group of second primary lung cancer. It should be noted that a peculiar affiliation between lung cancer/HER2-positive breast cancer and the interaction between the ER/EGFR pathways has been noted in previous studies [1,22]. Thirdly, whether the setting of the more-than-six-month intervals between the diagnosis dates of each cancer was appropriate or not remained doubtful. A previous study from Lin et al. only found the association in a manner of synchronicity, so we tried to investigate the association beyond six-month intervals. Moreover, we defined the diagnosis of breast cancer using the code (ICD-9-CM 174), which does not include ductal carcinoma in situ of the breast. The correlation between breast cancer and second primary lung cancer could be either strengthened or weakened with the addition of ductal carcinoma in situ of the breast to the breast cancer group. Finally, breast cancer patients might have a higher risk of incidental lung cancer diagnoses through routine follow-up imaging tests compared with non-cancer patients.

## 5. Conclusions

In conclusion, the key finding from this retrospective study is that patients with breast cancer had a significantly higher risk of developing second primary lung cancer compared with patients without breast cancer, particularly among younger and non-comorbid groups. The special association may be attributed to some potential risk factors such as inherited susceptibility and long-term exposure to PM2.5, and is supposed to increase public awareness. Moreover, the probability of developing second primary lung cancer should always be kept in mind and should encourage subsequent lung cancer screening on a regular basis for young Asian women with breast cancer. Further studies with more optimal designs in the future will be necessary, given the fact that inherited genotypes, different subtypes of breast cancer and lung cancer, and other unrecognized etiologies may play vital roles in both cancers’ development.

## Figures and Tables

**Figure 1 cancers-14-02977-f001:**
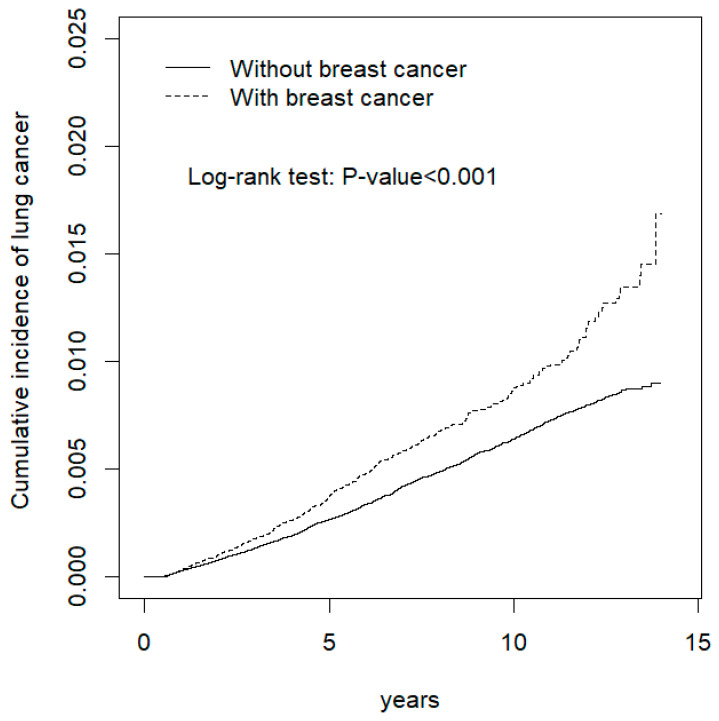
Cumulative incidence curves of lung cancer for the breast cancer and cancer-free healthy group.

**Table 1 cancers-14-02977-t001:** Demographic factors and comorbidities of the study participants according to breast cancer status.

	Age, and Index Year Matched					Propensity Score Matched				
	Control *N* = 377,804		Breast Cancer *N* = 94,451		*p*-Value *	Control *N* = 94,451		Breast Cancer *N* = 94,451		*p*-Value *
Variable	*N*	%	*N*	%		*N*	%	*N*	%	
Age, years					0.99					0.94
≤49	172,492	(45.7)	43,123	(45.7)		43,081	(45.6)	43,123	(45.7)	
50–64	144,924	(38.4)	36,231	(38.4)		36,219	(38.4)	36,231	(38.4)	
≥65	60,388	(16.0)	15,097	(16.0)		15,151	(16.0)	15,097	(16.0)	
Means (SD)	52.7	12.1	52.7	11.9	0.06	52.9	(12.1)	52.7	(12.0)	0.01
Urbanization ^†^					<0.001					0.001
1 (highest)	124,803	(33.0)	35,247	(37.3)		30,841	(32.7)	35,247	(37.3)	
2	113,256	(30.0)	28,273	(29.9)		28,505	(30.2)	28,273	(29.9)	
3	60,979	(16.1)	14,641	(15.5)		15,194	(16.1)	14,641	(15.5)	
4 (lowest)	78,766	(20.9)	16,290	(17.3)		19,911	(21.1)	16,290	(17.3)	
Comorbidity										
CAD	20,709	(5.48)	5077	(5.38)	0.2	5093	(5.39)	5077	(5.38)	0.87
Stroke	20,481	(5.42)	4546	(4.81)	<0.001	4559	(4.83)	4546	(4.81)	0.89
COPD	6482	(1.72)	1512	(1.60)	0.01	1519	(1.61)	1512	(1.60)	0.90
Hyperlipidemia	16,201	(4.29)	4692	(4.97)	<0.001	4686	(4.96)	4692	(4.97)	0.95
Hypertension	59,708	(15.8)	22,380	(23.7)	<0.001	22,421	(23.7)	22,380	(23.7)	0.82
Diabetes	34,080	(9.02)	12,258	(13.0)	<0.001	12,335	(13.1)	12,258	(13.0)	0.60
Tuberculosis	1299	(0.34)	304	(0.32)	0.30	252	(0.27)	304	(0.32)	0.03
Chronic kidney disease	1864	(0.49)	506	(0.54)	0.10	640	(0.68)	506	(0.54)	0.001
Chronic liver disease and Cirrhosis	7111	(1.88)	2068	(2.19)	<0.001	2132	(2.26)	2068	(2.19)	0.32

SD, standard deviation; CAD, coronary artery disease; COPD, chronic obstructive pulmonary disease. ^†^: The urbanization level was categorized by the population density of the residential area into 4 levels, with level 1 as the most urbanized and level 4 as the least urbanized. * Comparison between breast cancer and control.

**Table 2 cancers-14-02977-t002:** The incidence and risk factors for lung cancer.

	Event	PY	Rate ^#^	Crude HR (95% CI)	Adjusted HR ^&^ (95% CI)
Breast cancer					
No	1438	242,1761	5.94	1.00	1.00
Yes	447	545,413	8.20	1.41 (1.27, 1.57) ***	1.34 (1.20, 1.49) ***
Age, year					
20–49	419	1,465,317	2.86	1.00	1.00
50–64	826	1,089,142	7.58	2.74 (2.44, 3.08) ***	2.41 (2.13, 2.71) ***
≥65	640	412,715	15.5	5.76 (5.09, 6.51) ***	4.17 (3.63, 4.79) ***
Urbanization ^†^					
1 (highest)	619	1,005,478	6.16	1.00	1.00
2	549	890,551	6.16	1.00 (0.89, 1.12)	0.98 (0.87, 1.10)
3	280	472,245	5.93	0.96 (0.84, 1.11)	0.94 (0.82, 1.09)
4 (lowest)	437	598,900	7.30	1.18 (1.05, 1.34) **	0.98 (0.86, 1.11)
Comorbidity					
CAD					
No	1705	2,800,559	6.09	1.00	1.00
Yes	180	166,615	10.8	1.76 (1.51, 2.06) ***	0.68 (0.57, 0.81) ***
Stroke					
No	1678	2,809,136	5.97	1.00	1.00
Yes	207	158,038	13.1	2.19 (1.89, 2.53) ***	0.89 (0.76, 1.05)
COPD					
No	1781	2,917,837	6.10	1.00	1.00
Yes	104	49,337	21.1	3.46 (2.84, 4.22) ***	1.68 (1.37, 2.08) ***
Hyperlipidemia					
No	1713	2,832,027	6.05	1.00	1.00
Yes	172	135,147	12.7	2.09 (1.79, 2.44) ***	1.16 (0.97, 1.37)
Hypertension					
No	1119	2,443,908	4.58	1.00	1.00
Yes	766	523,266	14.6	8.18 (2.90, 3.49) ***	1.97 (1.76, 2.22) ***
Diabetes					
No	1507	2,675,199	5.63	1.00	1.00
Yes	378	291,975	13.0	2.29 (2.05, 2.57) ***	1.09 (0.95, 1.23)
Tuberculosis					
No	1875	2,959,327	6.34	1.00	1.00
Yes	10	7847	12.7	2.14 (1.15, 3.99) *	1.31 (0.70, 2.45)
Chronic kidney disease					
No	1878	2,956,303	6.35	1.00	1.00
Yes	7	10,871	6.44	1.10 (0.52, 2.30)	0.63 (0.30, 1.32)
Chronic liver disease and Cirrhosis					
No	1852	2,918,334	6.35	1.00	1.00
Yes	33	48,840	6.76	1.11 (0.79, 1.56)	0.63 (0.45, 0.90) **

CI, confidence interval; CAD, coronary artery disease; COPD, chronic obstructive pulmonary disease; HR, hazard ratio; PY, person-years; Rate ^#^, incidence rate per 10,000 person-years; ^&^ Multivariable analysis including age, urbanization level and comorbidities of CAD, stroke, COPD, hyperlipidemia, hypertension, diabetes, tuberculosis, chronic kidney disease, and chronic liver disease and cirrhosis; * *p* < 0.05, ** *p* < 0.01, *** *p* < 0.001.

**Table 3 cancers-14-02977-t003:** Incidence and hazard ratios of lung cancer between individuals with and without breast cancer.

	Breast Cancer							
	No			Yes				
	Event	PY	Rate ^#^	Event	PY	Rate ^#^	Crude HR (95% CI)	Adjusted HR ^&^ (95% CI)
Age, year								
20–49	283	1,194,014	2.37	136	271,304	5.01	2.18 (1.77, 2.67) ***	2.11 (1.72, 2.59) ***
50–64	628	890,350	7.05	198	198,792	9.96	1.46 (1.24, 1.71) ***	1.34 (1.14, 1.58) ***
≥65	527	337,397	15.6	113	75,318	15.0	0.99 (0.81, 1.21)	0.93 (0.75, 1.14)
Comorbidity ^§^								
No	619	1,871,255	3.31	225	380,331	5.92	1.81 (1.56, 2.11) ***	1.94 (1.66, 2.26) ***
Yes	819	550,506	14.9	222	165,082	13.5	0.95 (0.82, 1.10)	0.93 (0.80, 1.08)
Follow-up years								
<5	785	1,554,604	5.05	254	367,803	6.91	1.39 (1.20, 1.60) ***	1.84 (1.29, 2.11) ***
5–10	540	718,305	7.52	152	149,190	10.2	1.36 (1.13, 1.62) ***	1.31 (1.10, 1.57) ***
>10	113	148,850	7.59	41	24,820	14.4	1.91 (1.33, 2.73) ***	1.84 (1.28, 2.64) ***

CI, confidence interval; HR, hazard ratio; PY, person-years; Rate ^#^, incidence rate per 10,000 person-years; ^&^ Multivariable analysis including age, urbanization level and comorbidities of CAD, stroke, COPD, hyperlipidemia, hypertension, diabetes, tuberculosis, chronic kidney disease, and chronic liver disease and cirrhosis; ^§^ Individuals with any comorbidity of CAD, stroke, COPD, hyperlipidemia, hypertension, diabetes, tuberculosis, chronic kidney disease, and chronic liver disease and cirrhosis were classified into the comorbidity group; *** *p* < 0.001.

**Table 4 cancers-14-02977-t004:** Overall lung cancer incidence (per 1000 person-years) and estimated HRs in breast cancer patients compared with those without breast cancer in the propensity-score matching.

	Propensity Score Matched	
	Breast Cancer	
Variables	No (*N* = 94,451)	Yes (*N* = 94,451)
**Lung cancer**		
Person-years	604,590	545,413
Follow-up time (y), Mean ± SD	6.40 ± 3.66	5.77 ± 3.58
Event, n	422	447
Rate ^#^	6.98	8.2
cHR (95% CI)	1 (Reference)	1.20 (1.05, 1.37) **
aHR (95% CI) ^&^	1 (Reference)	1.21 (1.06, 1.38) **

CI, confidence interval; HR, hazard ratio; Rate ^#^, incidence rate per 10,000 person-years; ^&^ Multivariable analysis including age, urbanization level and comorbidities of CAD, stroke, COPD, hyperlipidemia, hypertension, diabetes, tuberculosis, chronic kidney disease, and chronic liver disease and cirrhosis; ** *p* < 0.01.

## Data Availability

Data are available in a publicly accessible repository that does not issue DOIs. Publicly available datasets were analyzed in this study.

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
