# Peer review of "Association between Breast Cancer and Second Primary Lung Cancer among the Female Population in Taiwan: A Nationwide Population-Based Cohort Study"

_cancers, 2022, doi:10.3390/cancers14122977_

Round 1

Reviewer 1 Report

The authors have addressed all the major comments.

However, the reported inverse associations between some comorbid conditions (e.g., CAD, stroke, CKD, liver disease) and risk of lung cancer may be confounded by unmeasured factors such as smoking. For example, smokers may quit smoking after diagnosis of the above disease conditions, lowering their risk of lung cancer. The authors claimed that the prevalence of smoking is very low in Taiwan, but there are no data to support this claim. This important explanation should be added to the revision and the authors should clearly acknowledge the limitation of not having information on smoking – the main risk factor for lung cancer – in the limitation section.

The manuscript can be accepted after implementing the above minor suggestions and checking all the sentence structures for clarity.

Reviewer 2 Report

Thank you very much for your nice revisions. Authors have addressed most of my methodology comments in a satisfactory way. However, the following should also be noted.

1. Please provide references of the newly added epidemiology in the Introduction.

2. For comments #2, #4, #5, I suggest putting your replies into your methodology.

3. Please provide the details of SIR analysis into your statistical analysis part

4. The study implications and articulations or applications were still under-discussed. This is a major concern. Suggest discussing these more in detailed.

Round 2

Reviewer 2 Report

Authors have addressed all of my concerns and I think the current form is suitable to be published in Cancers.

This manuscript is a resubmission of an earlier submission. The following is a list of the peer review reports and author responses from that submission.

Round 1

Reviewer 1 Report

Lin and colleagues investigated the association between breast cancer and risk of primary lung cancer among women in Taiwan using a nationwide cohort study. The authors found a positive association between breast cancer and risk of primary lung cancer, after controlling for several relevant factors using conventional covariate adjustment and propensity score methods. The results are not so surprising, and there are major methodological concerns in the paper, as highlighted below.

METHODS

  1. Line 83: It is established that patients with advanced cancer stage (e.g., stages III-IV) have a much higher risk of metastasis. Given this, breast cancer patients diagnosed at stages III/IV may have increased risk of metastasis (e.g., lung cancer), which may be difficult to be distinguished from primary lung cancer. To avoid this major bias, the authors should have used only breast cancer patients diagnosed at stages I-II. Comparing the risk of primary lung cancer between stage I-IV breast cancer patients and cancer-free controls is therefore not appropriate.
  2. Smoking is an established risk factor for lung cancer. Why did the authors not adjust for this important factor and also assess whether and to what extent the association is modified by smoking?
  3. Why did the authors not also consider renal and liver diseases as comorbidities in the study? These are also important comorbidities among breast cancer patients.

RESULTS

  1. Lines 167-169: The authors should indicate that the magnitude of the association was attenuated after controlling for potential confounding factors with the use of propensity score matching (from 34% to 22% increased risk). This should be discussed in the discussion section as well.

DISCUSSION

  1. There should be extensive explanation for why no associations were observed among older patients and those with significant comorbidities (Table 3).
  2. Any explanation for the inverse association between CAD and lung cancer risk (Table 2)?
  3. Lines 223-230: The authors claimed that the observed higher risk of primary lung cancer among breast cancer patients could be due to long-term exposure to fine particulate matter. Given that air pollution differs by geographical region, it would be of interest if the authors had included information on geographical region in the models and also assessed whether this variable modifies the reported association. This would certainly lend credence to their hypothesis.

GENERAL

  1. The manuscript needs scientific editing, especially the Abstract and Introduction sections.

Reviewer 2 Report

Thank you very much for your work. This is a national-wide population-based study with a very good study concept. The large sample size makes the study have a potential of great impact. However, I think the authors can consider the following to further enhance the robustness of the study as the current methodology is rather too simple and not enough to draw a valid conclusion. Points in bold are my major concerns.

  1. Introduction part is too simple. More descriptions and references are required to summarize the epidemiology, public health burden of the health problems.
  2. Line 48, 52. References are missing.
  3. Data source – suggest to describe more about the data source e.g. what hospitals are obligated to submit the information to the registry? Any incentives or registration fees are paid to the reporting hospitals? Have registry database been systematically converted to ICD-9 code and how?
  4. Are the registry database linked with death certificates from the National Health Database?
  5. For persons not identified by the above process, were they considered to be alive (passive follow-up)?
  6. Exclusion criteria – how about those with missing birth dates, missing follow-up or death status?
  7. Kindly include a flowchart of study population inclusion detailing the number of reasons of exclusion at each stage.
  8. Line 86. I am very confused with the purpose of matching here. Is it a propensity score matching? What are the matching criteria? Why is it a 1:4 ratio? How did they perform matching? Does the control group mean those without breast cancer only or without any cancer (so-called healthy cohort?)? I think matching is not required for the incidence estimation and authors have the full data of the population.
  9. A robust and common method to compare the incidence of lung cancer between breast cancer patients and the “general healthy cohort (no matching required)” is by standard incidence ratio (SIR) and absolute excessive risk (AER). Apart from SIR and AER in the general comparison, these figures in subgroups are important e.g. by age, follow-up period, year of diagnosis, radiotherapy given, smoking, etc. Authors should have all of the data from the registry and I think they should be able to do so. Authors may refer to two previously published papers in Taiwan and Hong Kong respectively, about the second malignancy of another solid tumour (Chen, et al. The incidence and risk of second primary cancers in patients with nasopharyngeal carcinoma: a population-based study in Taiwan over a 25-year period (1979-2003). Ann Oncol. 2008;19(6):1180-1186. doi:10.1093/annonc/mdn003) and (Chow, et al. Second primary cancer after intensity-modulated radiotherapy for nasopharyngeal carcinoma: A territory-wide study by HKNPCSG. Oral Oncol. 2020;111:105012. doi:10.1016/j.oraloncology.2020.105012).
  10. Suggest to move the part of “validity of Cancer Diagnoses in NHI Database” to section 2.1.
  11. After SIR and AER are demonstrated, I think the Cox proportional hazard model could focus on the breast cancer patients themselves to see what are the risk factors of lung cancer development AMONG the breast cancer patients (refer to Chow et al). Again, authors should also consider age, follow-up period, year of diagnosis, radiotherapy given, smoking, etc.
  12. Of course the authors may include your initially matched analyses as a supplementary while authors should explain the functions of matched analysis.
  13. Line 112. What is the purpose of propensity score matching here for sensitivity analysis? Is it the same as the matching in 1:4 ratio? Table 2 multivariate analysis has already shown the adjusted hazard ratios.
  14. Sensitivity analyses could be the truncated analyses by follow-up period. Follow-up period is certainly a risk factor for the development of second cancer and the risk may not be truly reflected as some patients have relatively short follow-up. Authors should select the suitable timepoint (e.g. median follow-up) for the truncation.
  15. Kindly update the cumulative incidence curve to show the situation in breast cancer patients and ALL of the persons without matching.
  16. Table 1 – I would like to see more clinical characteristics of the breast cancer cohort e.g. age, follow-up period, year of diagnosis (5-year), radiotherapy given, smoking, etc.
  17. Table 3—What is the reference of the hazard ratio? Is it a subgroup analysis of Table 2?
  18. Many secondary cancers occur after a latency following the first cancer. Therefore the longer the patient survives, the higher the risk for a second cancer becomes. I would like to see the Kaplan–Meier survival curves since breast cancer diagnosis for (1) breast cancer patients with second lung cancer and (2) breast cancer patients without further lung cancer, and see if there is cross-over of the curves and the long-term survival difference (e.g. 5-year; 10-year; or even 15-year).
  19. It would be interesting to include a survival curve and survival statistics of the breast cancer patients with lung cancer.
  20. Discussion Line 185 – Authors stated that younger patient group have a higher risk here but Table 2 hazard ratios for the older groups is much higher and the incidence of the elderly group is indeed higher in Table 3.
  21. Authors should also explain the findings related to age as well as the updated findings.
  22. Authors discussed extensively the possible reasons of the findings while the study implications are absent. They are suggested to discuss the study implications and any potential articulations e.g. screening, extra clinical cautions, policy-making, etc.

Round 2

Reviewer 1 Report

The authors addressed only a few of the comments and the ones that were not addressed are outlined below. Also, the authors should have added a response to the various comments from the reviewers.

METHOD

  1. It is established that patients with advanced cancer stage (e.g., stages III-IV) have a much higher risk of metastasis. Given this, breast cancer patients diagnosed at stages III/IV may have increased risk of metastasis (e.g., lung cancer), which may be difficult to be distinguished from primary lung cancer. To avoid this major bias, the authors should have used only breast cancer patients diagnosed at stages I-II. Comparing the risk of primary lung cancer between stage I-IV breast cancer patients and cancer-free controls is therefore not appropriate. This important comment was not addressed.
  2. Smoking is an established risk factor for lung cancer. Why did the authors not adjust for this important factor and also assess whether and to what extent the association is modified by smoking? This important comment was not addressed.
  3. Why did the authors not also consider renal and liver diseases as comorbidities in the study? These are also important comorbidities among breast cancer patients. This comment was not addressed.

RESULTS

  1. The authors should indicate that the magnitude of the association was attenuated after controlling for potential confounding factors with the use of propensity score matching (from 34% to 22% increased risk). This should be discussed in the discussion section as well. This was not addressed.

DISCUSSION

  1. Any explanation for the inverse association between CAD and lung cancer risk (Table 2)? This was not addressed.
  2. The authors claimed that the observed higher risk of primary lung cancer among breast cancer patients could be due to long-term exposure to fine particulate matter. Given that air pollution differs by geographical region, it would be of interest if the authors had included information on geographical region in the models and also assessed whether this variable modifies the reported association. This would certainly lend credence to their hypothesis. This was not addressed.

GENERAL

  1. The manuscript still needs scientific editing.